# Efficacy and Safety of Two-Drug Regimens That Are Approved from 2018 to 2022 for the Treatment of Human Immunodeficiency Virus (HIV) Disease and Its Opportunistic Infections

**DOI:** 10.3390/microorganisms11061451

**Published:** 2023-05-31

**Authors:** Palanisamy Sivanandy, Jess Ng Yujie, Kanini Chandirasekaran, Ooi Hong Seng, Nur Azrida Azhari Wasi

**Affiliations:** 1Department of Pharmacy Practice, School of Pharmacy, International Medical University, Kuala Lumpur 57000, Malaysia; 2School of Pharmacy, International Medical University, Kuala Lumpur 57000, Malaysia; 3Department of Pharmacy, University of Malaya Medical Centre, Kuala Lumpur 59100, Malaysia; azrida@ummc.edu.my

**Keywords:** HIV, immunodeficiency, treatment, immunocompromised, safety, efficacy, infection

## Abstract

The human immunodeficiency virus (HIV) is a type of virus that targets the body’s immune cells. HIV infection can be divided into three phases: acute HIV infection, chronic HIV infection, and acquired immunodeficiency syndrome (AIDS). HIV-infected people are immunosuppressed and at risk of developing opportunistic infections such as pneumonia, tuberculosis, candidiasis, toxoplasmosis, and Salmonella infection. The two types of HIV are known as HIV-1 and HIV-2. HIV-1 is the predominant and more common cause of AIDS worldwide, with an estimated 38 million people living with HIV-1 while an estimated 1 to 2 million people live with HIV-2. No effective cures are currently available for HIV infection. Current treatments emphasise the drug’s safety and tolerability, as lifelong management is needed to manage HIV infection. The goal of this review is to study the efficacy and safety of newly approved drugs from 2018 to 2022 for the treatment of HIV by the United States Food and Drug Administration (US-FDA). The drugs included Cabotegravir and Rilpivirine, Fostemsavir, Doravirine, and Ibalizumab. From the review, switching to doravirine/lamivudine/tenofovir disoproxil fumarate (DOR/3TC/TDF) was shown to be noninferior to the continuation of the previous regimen, efavirenz/emtricitabine/tenofovir disoproxil fumarate (EFV/FTC/TDF) in virologically suppressed adults with HIV-1. However, DOR/3TC/TDF had shown a preferable safety profile with lower discontinuations due to adverse events (AEs), lower neuropsychiatric AEs, and a preferable lipid profile. Ibalizumab was also safe, well tolerated, and had been proven effective against multiple drug-resistant strains of viruses.

## 1. Introduction

Human immunodeficiency virus (HIV) infections remain one of the major public health concerns nowadays. HIV is a type of virus that targets the cells that are responsible for the body’s immune system. It can be transmitted through contact with the body fluids of HIV-infected people, especially during unprotected sexual intercourse or through the sharing of injection equipment. People with HIV infection will experience flu-like symptoms in stage 1, acute HIV infection. If HIV infections are left untreated, they will progress into stage two which is chronic HIV infection, and eventually develop into the most severe stage of HIV infection, known as acquired immunodeficiency syndrome (AIDS) [1]. People affected by HIV infection are generally immunosuppressed and become more vulnerable to other infections and diseases. HIV-infected people possess a high risk of developing opportunistic infections due to their weakened immune systems. The opportunistic infections that are commonly seen in HIV-infected people include pneumonia, tuberculosis, candidiasis, toxoplasmosis, and Salmonella infection [2,3]. These opportunistic infections can be managed with their corresponding treatments.

HIV infections are divided into two types: HIV type 1 (HIV-1) and HIV type 2 (HIV-2). HIV-1 is the predominant cause of AIDS worldwide, with an estimated 38 million people living with HIV-1. Meanwhile, HIV-2 is most likely to be found in West Africa, with an estimation of 1 to 2 million people living with HIV-2 [4]. HIV-1 and HIV-2 can be recognized because HIV-2 is less infectious. HIV-2 also comes with a lower viral load in the body and the progression to develop opportunistic infections is much slower. It is important to distinguish between HIV-1 and HIV-2 during diagnosis to allow for choosing the appropriate types of treatment, as some of the antiretroviral drugs are not effective against HIV-2 infection [4]. Currently, with our understanding, no effective cure for HIV infections has been developed yet. However, there are some effective treatments and management options available for HIV infection. More than 30 different anti-HIV drugs with different mechanisms of action have been approved for clinical use in HIV infections to drive viral loads down until an undetectable level [5]. Since HIV infections cannot be completely eradicated for now, lifelong treatment is needed to manage HIV infections. Therefore, current treatments have shifted to focus on the safety and tolerability of the drugs [6,7]. Combination antiretroviral therapy has introduced single-pill combination daily regimens to increase people’s adherence to HIV infection therapy [7,8]. People with HIV infection that is managed with proper medical care can live an increased, healthy lifespan and protect others from being infected with HIV [1]. Drugs such as Cabotegravir and Rilpivirine, Fostemsavir, Doravirine, and Ibalizumab were developed and approved from 2018 to 2022 for the treatment of HIV. This review is conducted to study the efficacy and safety of Cabotegravir and Rilpivirine, Fostemsavir, Doravirine, and Ibalizumab which are used in the treatment of HIV.

## 2. Materials and Methods

A systematic review contains the most up-to-date and comprehensive information, and it serves as an important tool for healthcare professionals to assess the reliability and clinical significance of a topic or intervention. A systematic review can convey summarised information and includes a comprehensive and systematic search of the literature, as well as reducing any selection bias that may be found in a review. It includes a synthesis of previous research results, which can be accompanied by a meta-analysis, which entails analysing the data and combining it into an average result. After that, the synthesis is used to draw conclusions and make recommendations.

### 2.1. Outcome Assessment

The primary outcomes of this review were the efficacy and safety of two-drug regimens that were approved from 2018 to 2022 for the treatment of human immunodeficiency virus (HIV) disease and its opportunistic infections. The efficacy of anti-HIV drugs, measured by the reduction in HIV-RNA and the elevation of CD4 cell count, was considered for this review. Drug safety includes the reduction in adverse drug effects and adverse events compared with previously introduced drugs on the market that had been approved by the US FDA.

### 2.2. Data Extraction

The data extracted for this review focuses mainly on the two-drug regimens used in the treatment of HIV infections. Preferred reporting items for systematic reviews and meta-analysis (PRISMA) were used for the selection of articles and reporting of reviews that evaluated randomised controlled trials. The main sources of data used were PubMed, NVivo, Mendeley, Evernote, CiteUlike, Biohunter, Delve health, Scicurve, Google Scholar, etc. Articles on nonretroviral medications, three- or four-drug regimens, and animal studies were excluded. The anti-HIV drugs included in our studies are those that were approved by the US FDA from 2018 to 2022, according to Centre Watch. 

There were altogether four drugs approved in the search period that were reviewed, and 13 articles of randomised controlled trials (RCTs) were collected. The details of data extraction are presented in Figure 1. The four drugs approved during the search period are Cabenuva (a combination of extended-release injectable suspension of cabotegravir and rilpivirine) in 2021, Rukobia (an extended-release tablet of fostemsavir) in 2020, Pifeltro (tablet doravirine), and Trogarzo (a parenteral formulation of ibalizumab-uiyk) in 2018 by the US-FDA. These drugs are administered in combination with other antiretroviral(s), for the treatment of HIV infections. All the authors independently extracted the relevant information from the RCT studies that fulfilled our inclusion criteria and any disagreements were resolved by consensus. The information extracted included the trial phase, region, conditions of subjects (viral load, CD4 count, and opportunistic infections), and outcome measures. This information was gathered and summarised into paragraphs, introducing each anti-HIV drug comprehensively.

## 3. Results

The results of this systematic review are summarised in three sections: the mechanism of action of the drugs, efficacy in treating HIV and its opportunistic infections, and the safety profile of the HIV drugs. The details are summarised in the following sections.

### 3.1. Mechanism of Action

The following section describes the mechanism of action of drugs in treating HIV.

#### 3.1.1. Cabotegravir and Rilpivirine (Cabenuva)

Cabenuva is an antiretroviral (ARV) drug used to treat HIV-1 infection, especially among adult HIV patients with a suppressed viral load and who are virologically stable [9]. It is a long-lasting extended release that can be administered through injections. A complete dose involves administering two gluteal intramuscular injections of rilpivirine and cabotegravir, the main components of Cabenuva. Rilpivirine and Cabotegravir administration is available in the form of monthly injections. Cabotegravir is an integrase strand transfer inhibitor (INSTI) that inhibits the integrase enzyme, a crucial component of the retroviral replication process, prohibiting the integration of viral DNA into the genome of the host cell [10]. Rilpivirine is a non-nucleoside reverse transcriptase inhibitor (NNRTI) that blocks the reverse transcriptase enzyme from transforming viral RNA into DNA. The flexibility of rilpivirine's structure around its aromatic rings enables it to adapt to modifications in the non-nucleoside RT binding pocket, which lowers the possibility of viral mutations resulting in resistance. Cabenuva combines INSTI Cabotegravir with NNRTI Rilpivirine, where Cabotegravir inhibits HIV replication, which is crucial in the virus’s replication cycle and establishes chronic infections, while Rilpivirine interferes with the reverse transcriptase enzyme. Orkin et al. [11] state that by blocking the function of these enzymes, Cabotegravir and Rilpivirine work together to prevent virus replication and reduce the viral load in the body. This helps to reduce the spread of HIV and improve the patient’s immune system. Cabotegravir is primarily metabolised by UGT1A1, with help from UGT1A9. Drugs that strongly induce UGT1A1 or UGT1A9 are expected to lower cabotegravir plasma concentrations and result in a loss of virologic response; therefore, coadministration of Cabenuva with these drugs is contraindicated.

#### 3.1.2. Fostemsavir (Rukobia)

Fostemsavir (Rukobia) is an HIV type-1 gp120-directed attachment inhibitor. It is used in combination with other antiretroviral(s) for the treatment of HIV-1 infection, mainly used in heavily treatment-experienced adults with multidrug-resistant HIV-1 infection who are failing their current ARV regimen due to resistance, intolerance, or safety concerns. This drug has been specifically used for HIV-1 patients who have undergone multiple treatments [12]. It is a prodrug of Temsavir (TMR), a fusion inhibitor that blocks HIV from entering immune cells by inhibiting the fusion of the infected cells and cellular membranes. The drug is metabolised primarily through hydrolysis (36.1%), oxidation (CYP3A4), and uridine diphosphate glucuronosyl transferases (UGT) (1%). According to Muccini [13], Rukobia was authorised as an oral, twice-daily extended-release tablet (600 mg). It has increased water solubility and acidic stability and is quickly converted to TMR, the only molecule detectable in plasma. It interacts with and suppresses the action of glycoprotein 120 (gp120), a subunit of the HIV-1 glycoprotein 160 (gp160) envelope glycoprotein that facilitates HIV-1 attachment to host-cell receptors and other immune cells, which is the first step towards preventing HIV-1 virus infection [12]. According to Kozal et al. [12], Rukobia is developed specifically for HIV-1 tropism, and it does not cross-resist with other kinds of ARV medications, including entry blockers such as ibalizumab, maraviroc, etc. Fostemsavir is a revolutionary ARV medicine with no cross-resistance to other ARVs and it provides an important alternative therapy option for people with multidrug-resistant HIV-1. Targeting gp120 subunits is a new and unique therapeutic strategy for HIV-1 infection therapy. Adding attachment inhibitors, such as TMR, to HIV-1 medications fills a critical need for therapeutic alternatives for patients with few effective options. By preventing the virus from entering human cells, Fostemsavir stops HIV replication and reduces the amount of virus in the body. Rukobia is contraindicated in patients who have previously developed hypersensitivity to fostemsavir or any of the components of Rukobia, as significant decreases in temsavir (the active moiety of fostemsavir) plasma concentrations may occur, potentially leading to loss of virologic response.

#### 3.1.3. Doravirine

Doravirine acts as a non-nucleoside reverse transcriptase inhibitor (NNRTI) [14]. The mode of action is to bind to the reverse transcriptase (RT) enzyme in HIV-1, which is required for the replication of the virus. HIV, with the aid of reverse transcriptase, is able to generate complementary DNA (cDNA) for the RNA genome [15]. Doravirine causes disruption by blocking RNA-dependent and DNA-dependent DNA polymerase activities at the active site of the enzyme [14]. Approximately 12 nM is 50% of the maximum inhibitory concentrations (EC50) for wild-type (WT) laboratory-adapted strains of HIV-1 [14]. Other enzymes, such as the human cellular DNA polymerase alpha (α), beta (β), and mitochondrial gamma (γ) will not be affected by Doravirine [14,15].

The in vitro-associated mutations of Doravirine are V106A, V106I, V106M, V108I, H221Y, F227C, F227I, F227L, F227V, M230I, L234I, P236L, and Y318F [16]. Based on the clinical trials, 36% of subjects presented with Doravirine-associated resistance substitutions in RT, whereas 28% of subjects showed genotypic and phenotypic resistance towards other antiretroviral drugs [17]. Doravirine is primarily eliminated through the cytochrome P450 3A4/5 pathway. Only 6% of a given dose is recovered unchanged in the urine, with even less recovered in the faeces. Doravirine should not be used in conjunction with CYP450 enzyme inducers because the drugs may lower the serum concentration of Doravirine and reduce its efficacy.

#### 3.1.4. Ibalizumab-Uiyk

Ibalizumab is an entry inhibitor that acts as a recombinant humanised immunoglobulin (Ig) G4 Mab in which it selectively binds to the amino acid site, especially at E77 and S79 on domain 1, whereas L96, P121, P122, and Q163 on domain 2 of the CD4^+^ T-cell receptors [18]. The binding process starts when HIV is attached to the primary CD4 receptors of T helper cells [18]. Variable loops of V1 to V5 are present in the gp120 (trimeric molecule) domain that mainly act as major receptor-binding sites. This conformational shift transforms the V1, V2, and V3 from a closed to an open state [18]. The HIV fusion entry can be prevented by the conformational changes in the CD4^+^ T-cell receptor and gp120 complex caused by ibalizumab, as it increases the steric hindrance [18]. This leads to the inhibition of interaction in gp120 against the HIV coreceptors CCR5 and CXCR4 (tropic strains) via the V3 loop [18]. Ibalizumab acts broadly on tropic strains by targeting HIV-1 entry before the coreceptor and fusion processes occur which is known as a postattachment inhibitor [18,19].

In summary, Ibalizumab is a humanized monoclonal antibody that binds to the CD4 receptor on human T cells. CD4 is a protein present on the surface of these cells and acts as the primary receptor for HIV-1. By binding to the CD4 receptor, ibalizumab blocks the binding of the HIV-1 virus to the CD4 receptor, preventing the initial attachment of the virus to the host cell. Ibalizumab not only blocks the direct binding of HIV-1 to CD4, but it also induces allosteric structural changes in the CD4 receptor. This conformational change makes it more difficult for the HIV-1 virus to bind to other co-receptors (e.g., CCR5 or CXCR4) necessary for viral entry into the host cell. By interfering with the entry of HIV-1 into T cells, ibalizumab effectively inhibits viral replication. It prevents the virus from entering the target cells and spreading the infection further.

Since the binding sites of the major histocompatibility complex (MHC) class II molecules are far apart from the cellular epitope targeted by ibalizumab on CD4 receptors do not cause immunosuppression [20]. Ibalizumab has a low affinity for complement component 1q (C1q) as well as antibody-dependent cellular toxicity (ADCC) in the Fcɣ receptor I receptors (FcɣRI) of natural killer cells (NK) [20]. When there is a change in the amino acid of Phe instead of Leu 34 in the genetic sequencing of Leu 234-Leu 235-Gly 236-Gly 237-Pro 238-Ser 239, IgG4 experiences either weaker or no affinity [20]. Ibalizumab is metabolized by CD4 receptor internalisation and has no effect on liver or kidney metabolism. No major contraindications were reported for the concurrent administration of ibalizumab with other drugs.

### 3.2. Efficacy

The efficacy of drugs in the treatment of HIV is described in this section.

#### 3.2.1. Cabotegravir and Rilpivirine (Cabenuva)

Cabenuva is a combination of two ARV medications, Cabotegravir and Rilpivirine, indicated for the treatment of HIV-1 infection in adults with the virus that is suppressed on a stable ARV regimen. ViiV Healthcare conducted 12-month research on the efficacy of Cabenuva. Cabenuva might be successfully adopted across diverse healthcare settings in the United States, according to the clinical trial, which involved people living with HIV and healthcare teams [21]. Cabenuva was deemed adequate and acceptable by the HIV patients who took part in the experiment, with 97% expressing interest in receiving a long-acting medication instead of daily oral medication. The medical study also revealed that all individuals with an available viral load sustained viral suppression with no virological setbacks. In 72% of the individuals, the most prevalent overall adverse event was injection site hypersensitivity [21]. Smith [22] indicates that Cabenuva has demonstrated long-term effectiveness and tolerance as a sustaining medication for HIV-1 infection. At the 256th week, 81% of the participants in the randomised eight-week and four-week dosed groups and 93% of the participants who switched to their choice of dosing groups in the extension period, maintained an HIV-1 RNA level less than the required level of 50/mL. In addition, after the 48th week, there were no virologic failures as defined by the protocol. In the study, a long-acting injectable Cabenuva dose showed high efficacy with daily antiretroviral treatment through ninety-six weeks in HIV-1-infected naïve adults undergoing ARV therapy [22]. It is important to note that individual results may vary, and it is recommended to speak with a healthcare professional about the potential benefits and risks of Cabenuva.

#### 3.2.2. Fostemsavir (Rukobia)

Fostemsavir is a novel antiviral drug developed for the treatment of multidrug-resistant HIV-1 infections. Temsavir, the active moiety of Fostemsavir, binds directly with the glycoprotein (gp) 120 subunit in HIV-1 envelope gp160; this selectively inhibits the interaction between the virus and cellular CD4 receptors. In the final stages of its trial, Fostemsavir has been studied to address the need for novel treatment regimens, particularly for patients who have been intensively treated and have few alternatives left. Muccini et al. [22] studied the efficacy of Fostemsavir in one phase of the third trial involving 372 participants living with HIV-1 infection, failing the existing ARV treatment with multiple drug-resistant HIV infections; they found the drug effective in increasing CD4^+^ T-cells and decreasing HIV RNA. Kozal et al. [21] identified that at the 24th week, 53% of patients in the randomised group and 37% in a nonrandomised group achieved virologic suppression, followed by 54% and 38% at the 48th week and 60% and 37% at the 96th week. In interventional studies, Fostemsavir was shown to effectively lower the viral load in HIV-positive individuals with limited treatment options, including those who have developed resistance to multiple ARV drugs. These trials demonstrated that the combination of Fostemsavir with other ARV drugs dramatically lowered viral load and improved immune system function. Fostemsavir got approval from the US Food and Drug Administration (FDA) in 2021 and is now commercially available as a treatment option for HIV-1 infections.

#### 3.2.3. Doravirine

Doravirine (DOR) is commonly used in combination with lamivudine and tenofovir disoproxil fumarate (DOR/3TC/TDF). DOR/3TC/TDF causes a genotypically associated NNRTI resistance mutation (V1061) in a participant due to the susceptibility of the virus. The study reported a differential efficacy response for subgroups of participants aged 31 or older (median age) at week 48; however, this trend was not observed at week 96. Participants with high baseline HIV-1 RNA levels (>100,000 copies/mL) in the DOR/3TC/TDF group achieved HIV-1 RNA levels of 50 copies/mL at week 96 (treatment difference, 8.1%; 95% CI, 22.9% to 6.7%) [23]. A randomised double-blind study between the combination of DOR and Islatravir (ISL) shows higher efficacy in week 96 as the participants present with HIV-1 RNA levels of <50 copies per millilitre during the treatment regimen. Through week 96, both the islatravir-based regimens and the DOR/3TC/TDF regimen had similar overall safety and tolerability profiles. Throughout the trial, participants tolerated doravirine in combination with islatravir well [24].

#### 3.2.4. Ibalizumab-Uiyk

The US Food and Drug Administration (FDA) approved ibalizumab (IBZ) with the combination of other antiretroviral drugs (ART) used in multidrug resistance for HIV-1 patients [25]. IBZ doses of 10 and 15 mg/kg resulted in a significant increase in the mean CD4^+^ T-cell count compared to the placebo. Ibalizumab treatment reduced viral load at week 24 (−0.77 log_10_ arm A and −1.19 log_10_ arm B) and 48 weeks (−0.54 log_10_ and −0.77 log_10_) more than with a placebo. Compared with a placebo, there were statistically significant differences in viral load at week 24 (*p* = 0.001) and week 48 (*p* = 0.027). CD4^+^ T-cell counts increased significantly in both arms A and B compared to placebo by week 48. The efficacy of IBZ has been proven regardless of the viral tropism of multidrug-resistant strains [25].

### 3.3. Safety

The safety of the anti-HIV drugs is summarized in this section, emphasizing the safety profile of individual drugs.

#### 3.3.1. Cabotegravir and Rilpivirine (Cabenuva)

Cabenuva’s safety and acceptability have been thoroughly investigated in rigorous clinical trial programs such as the Antiretroviral Therapy as Long-Acting Suppression (ATLAS) study, First Long-Acting Injectable Regimen (FLAIR) trial, and the Antiretroviral Therapy as Long-Acting Suppression Every 2 Months (ATLAS-2M) study. Based on the findings of these studies [26], it is not safe to use Cabenuva in patients with previously hypersensitive reactions to rilpivirine or cabotegravir. The drug should also not be used by patients receiving medicines such as systemic dexamethasone, rifapentine, phenobarbital, rifabutin, oxcarbazepine, phenytoin, carbamazepine, St. John’s wort, and rifampin [26]. During postmarketing surveillance, Cabenuva with rilpivirine-containing treatments produced hypersensitivity reactions, including drug-reaction instances. While some skin responses are related to constitutional symptoms such as fever, others are connected to organ issues and problems such as biochemical increases in hepatic serum levels. If indications or symptoms of hypersensitivity responses appear, patients must stop taking Cabenuva immediately. Furthermore, clinical conditions, especially liver transaminases, should be examined, and necessary treatment should be started. Orally induced rilpivirine and cabotegravir may assist in identifying individuals at risk of hypersensitivity [26]. Severe postinjection responses were observed in less than 1% of participants shortly after receiving rilpivirine [26]. When formulating and injecting Cabenuva, carers must carefully comply with the instructions for use. The drug formulations should be administered slowly through intramuscular injection, and patients should be monitored for a few minutes after the injection to avoid unintentional parenteral injection. Individuals with an underlying liver illness before therapy may be at a higher risk of deteriorating. Monitoring liver chemistry is advised, and the medicine should be stopped if hepatotoxicity is detected. Cabenuva is a comprehensive HIV-1 therapy for adults and children over 12 who weigh at least 35 kg [26]. This treatment replaces the conventional ARV prescription for virologically suppressed individuals using stable ARV regimens with no history of therapeutic failure or known sensitivity to rilpivirine or cabotegravir.

#### 3.3.2. Fostemsavir (Rukobia)

Fostemsavir has high effectiveness and a low risk of side effects. However, because CYP3A4 plays a role in the drug’s metabolism, doctors should be mindful of potential drug reactions [22]. Rukobia [27] does not recommend using Fostemsavir in patients sensitive to Rukobia or any component used in its manufacture. Patients using medicines such as enzalutamide, carbamazepine, phenytoin, rifampin, mitotane, and hypericum perforatum should also avoid using the drug. Rukobia can have major side effects such as immune system alterations, heart rhythm abnormalities, and liver blood-test outcomes. The most prevalent Fostemsavir adverse effect is nausea [27]. Therefore, patients should inform their doctors about their medical conditions before taking Rukobia. Patients should disclose their heart conditions, liver problems, pregnancy status, and breastfeeding plans. It is unsafe to breastfeed when taking Rukobia when infected with HIV-1 due to the risk of transmission to newborns. Patients must also keep a list of the drugs they take and show it to their healthcare providers and pharmacists when using new medicines. This will help the healthcare providers ascertain if taking Fostemsavir with the other medicines is safe. According to Rukobia [27], patients should take Fostemsavir tablets as whole tablets without chewing, crushing, or splitting before swallowing.

#### 3.3.3. Doravirine

DOR/3TC/TDF shows a better safety profile with fewer neuropsychiatric adverse effects of 26.4% compared to 58.5% in the Efavirenz/Emtricitabine/Tenofovir (EFV/FTC/TDF) group based on the double-blind trial. The DOR/3TC/TDF group has favourable renal and lipid profiles [23]. A DOR of 100 mg is well tolerated regardless of gender [28]. In comparison with Efavirenz, DOR presents fewer adverse effects on the central nervous system (CNS) when observed in weeks 8 and 24 [29]. Based on the phase 2b trial, the combination of DOR and ISL did not significantly present with drug-related adverse effects from weeks 48 to 96 [24]. Thus, based on the trials conducted, a DOR of 100 mg can be used as an initial treatment for adults with HIV-1 since it has promising results [29].

#### 3.3.4. Ibalizumab-Uiyk

The combination of IBZ and ART was well tolerated as compared to placebo groups. The most common adverse effects experienced by patients with IBZ include headache, diarrhoea, nausea, somnolence, and a mild rash in around ≤ 10% of the participants. There were no significant adverse effects of immunogenicity or hepatotoxicity, unlike other ART drugs [25].

### 3.4. Comparison of the Efficacy and Safety of Newly Approved HIV Drugs from 2018 to 2022

The efficacy of Cabotegrevir and Rilpivirine is measured with the primary outcome, which is the proportion of participants with plasma HIV-1 RNA levels < 50 copies/mL at week 48. Noninferiority criteria at week 48 were met for the primary outcome. The analysis demonstrates that monthly injections of Cabotegrevir and Rilpivirine were noninferior to the daily oral current antiviral regimen: 600 mg abacavir, 50 mg dolutegravir, and 300 mg lamivudine for maintaining HIV-1 suppression. As for fostemsavir, the response rate remained similar between fostemsavir and ritonavir-boosted atazanavir (ATV/r) in HIV-1 treatment in this randomised, phase 2b, dose-finding study. The other approved drug is doravirine. In the DRIVE-AHEAD study, the noninferior efficacy of doravirine/lamivudine/tenofovir disoproxil fumarate relative to efavirenz/emtricitabine/tenofovir disoproxil fumarate was maintained through week 96. However, DOR/3TC/TDF was shown to be safer for use than EFV/FTC/TDF with fewer adverse effects. In the DRIVE-SHIFT study, it is also stated that switching to DOR/3TC/TDF had no significant benefit over the continuation of the previous regimen in virologically suppressed adults with HIV-1 but was associated with a favourable lipid profile. Therefore, switching to one dose daily of DOR, 3TC, or TDF is rather effective and provides a better-tolerated choice for suppression of the HIV virus in patients who are considering a therapy change. The last drug, ibalizumab at 15 mg/kg q2wk has a good safety profile, is well tolerated, and results in long-acting antiviral properties and an immunological response over 48 weeks. Ibalizumab is the most suitable option to be used in individuals with restricted treatment options due to the development of multidrug resistance as it is effective in viral inhibition, tolerability, and long-acting dosing compared to placebo.

In terms of safety, doravirine and ibalizumab are well tolerated and show a better safety profile. The common adverse events included dizziness, abnormal dreams, nausea, rash, headache, and vomiting. Using doravirine can cause nasopharyngitis but the effect is mild. As for Cabotegrevir and Rilpivirine, they can cause upper respiratory tract infection, diarrhoea, and headache. Most of the patients also complain about injection-site pain. As for fostemsavir, the primary adverse events are dizziness, headache, diarrhoea, and anxiety. The summary of Cabotegravir and Rilpivirine, Fostemsavir, Doravirine, and Ibalizumab with their efficacy and safety are shown in Table 1.

### 3.5. Nonpharmacological Management

In addition to pharmacological treatments, some nonpharmacological management can be used to relieve the symptoms of HIV. Based on the research, a few nonpharmacological treatments have proven their efficacy in improving the symptoms of HIV. Aerobic exercise and progressive resistance exercises are effective in reducing the pain caused by HIV. Regular physical activity has a protective anti-inflammatory effect, which is beneficial for HIV-related chronic inflammation. Similar to other chronic diseases, the improvement of muscle mass is important for HIV-infected people to preserve their muscle tropism and functional status [36,37]. Acupuncture has shown immunomodulatory effects and can improve symptoms in HIV-infected people with inflammatory conditions. Acupuncture works by helping relieve pain in the body by stimulating the release of chemicals. The use of acupuncture to target cholinergic anti-inflammatory pathways may inhibit the release of proinflammatory cytokines, thus reducing cardiovascular risk [36,38,39]. In addition, body therapies such as yoga and massage aid in reducing pain for some people. The level of immune cells attacked by HIV, which are CD4 cells, can be elevated by doing these body therapies. Yoga and massage can also reduce a patient’s anxiety and depression [38]. Lifestyle change regarding diets also brings benefits to HIV-infected people. HIV-infected people should always ensure that they have enough nutrition and nourishment. Healthy foods such as fruit, vegetables, whole grains, and lean protein can boost the immune system and thus maintain body strength. In addition, food that is raw and unsanitised should be avoided, as these foods may contain parasites that will cause serious infections in HIV-infected people who have weak immune systems [39].

Supplements and herbal medications are also potential management options for HIV-infected people. However, supplements and herbal medications should be used carefully, as some of them lack enough evidence to prove their efficacy and safety and might contraindicate HIV treatment. Patients should confirm and discuss with their respective healthcare providers before starting any supplements or herbal medication [38]. Loss of appetite is a common symptom that can be experienced by HIV-infected people, as some anti-HIV medications can upset people’s stomachs. Marijuana is effective in reducing pain, controlling nausea, and increasing people’s appetites. However, marijuana is only legal in a few countries. Herbal medications such as milk thistle are normally used to improve liver function, and it does not interact with anti-HIV medications [38]. Some of the supplements are known to be contraindicated with HIV medications and decrease the effectiveness of HIV treatment. One of them is a garlic supplement, which can decrease the effectiveness of medication when taken together. Although garlic can strengthen immune systems, the side effects still outweigh the benefits. Other than garlic supplements, St. John’s wort, a well-known depression remedy, is also known to reduce the effectiveness of several types of anti-HIV medications. The same goes for both echinacea and ginseng, as they also interact with anti-HIV medications [38,40].

## 4. Conclusions

In summary, this review introduces the efficacy and safety of new drugs approved from 2018 to 2022 for the treatment of HIV. After numerous comparisons of the article with different trials and reviews, it was found that ibalizumab has great efficacy and a good safety profile when used as a therapy for managing HIV either alone or in combination with other antiretroviral drugs Doravirine has noninferior efficacy compared to other drugs but has demonstrated a more favourable safety profile. The common adverse events associated with both drugs included dizziness, abnormal dreams, nausea, rash, headache, and vomiting. Further studies and trials might be needed to ensure and confirm their efficacy and safety profiles in the long term.

## Figures and Tables

**Figure 1 microorganisms-11-01451-f001:**
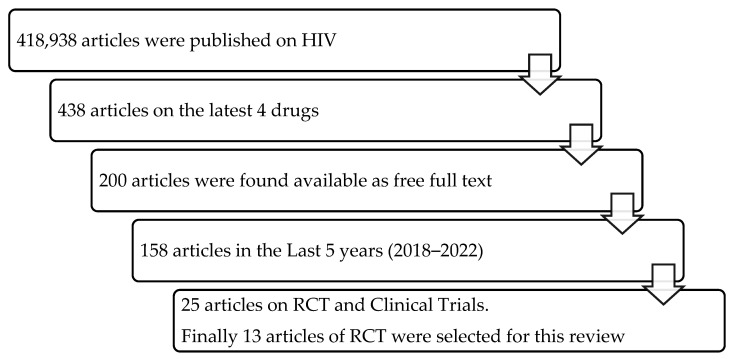
Schematic diagram of the data extraction for the review.

**Table 1 microorganisms-11-01451-t001:** Efficacy and safety of the newly approved drugs for the treatment of HIV.

Drug Name	Author, Year, Reference Number	Study Design	Population Characteristics	Interventions	Primary Outcome Measured	Efficacy	Safety
Cabotegrevir and Rilpivirine	Rizzardini G., et al., 2020 [30]	Randomized, open-label phase 3 studies	Patients aged ≥18 years old with virologically suppression, (HIV-1 RNA < 50 copies/mL)	Cabotegravir 400 mg, Rilpivirine 600 mg	Plasma HIV-1 RNA levels < 50 copies/mL at week 48	Participants had HIV-1 RNA ≥ 50 copies/mL at week 48, with an adjusted treatment difference of 0.16%, meeting the noninferiority criteria of the primary endpoint. Consistent efficacy can be seen between subgroups.	Major AEs: Nasopharyngitis, Headache, upper respiratory tract infection, diarrhoea, and back pain
Fostemsavir	Lagishetty C., et al., 2020 [31]	Randomized, double-blind, two-part, phase 1 study	Adults aged 18–49 years, body mass index of 18–32 kg/m^2^,	Fostemsavir 2400 BID or Fostemsavir 1200 QID or placebo BID	Difference of placebo in change for 7 days in the Fridericia-corrected QT interval (ddQTcF)	Fostemsavir 1200 mg b.i.d. did not result in a clinically meaningful change.	Major AEs: Dizziness, headache, and diarrhoea
	Lataillade M., et al., 2018 [32]	Randomized, active-controlled, partially blinded, phase 2b trial	Plasma HIV-1 RNA ≥ 1000 copies/mL, CD4^+^ T-cell count > 50 cells/mm^3^	Fostemsavir 400 mg BID, Fostemsavir 800 mg BID, Fostemsavir 600 mg QD, Fostemsavir 1200 mg QD	HIV-1 RNA < 50 copies/mL at week 24	The proportion of participants achieving HIV-1 RNA < 50 copies/mL was the same between the fostemsavir and ritonavir-boosted atazanavir	Major AEs: -
	Gartland M., et al., 2018 [33]	Randomized, placebo-controlled, double-blind, phase 3 clinical trial	Patients aged ≥18 years, plasma HIV-1 RNA ≥ 400 copies/mL, failing antiretroviral regimen and have no effect on the currently approved drug	Fostemsavir 600 mg BID, placebo BID	HIV-1 RNA < 40 copies/mL at Week 96	163/272 (60%) of participants (with 1 or 2 remaining approved fully active antiretrovirals) achieved HIV-1 RNA < 40 copies/mL at Week 96	Major AEs: -
Doravirine	Orkin C., et al., 2021 [34]	Randomized, active-controlled, double-blind, phase 3 trial	Adults with plasma HIV-1 RNA levels ≥1000 copies/mL	Doravirine 100 mg, Lamivudine 300 mg, TDF 300 mg or Efavirenz 600 mg, Emtricitabine 200 mg, TDF 300 mg	HIV-1 RNA levels < 50 copies/mL at week 96	HIV-1 RNA < 50 copies/mL was achieved by 77.5% of Doravirine/Lamivudine/TDF vs. 73.6% of Efavirenz/Emtricitabine/TDF participants, with a treatment difference of 3.8% at week 96. Noninferior efficacy of DOR/3TC/TDF seen through week 96.	Major AEs: Dizziness, abnormal dreams, nausea, and rash
	Molina JM., et al., 2021 [23]	Randomized, double-blind, dose-ranging trial	Adults aged ≥18 years, HIV-1 infection, clinically stable	Islatravir 0.25 mg, 0.75 mg, 2.25 mg doravirine 100 mg and fixed-dose DOR/3TC/TDF	HIV-1 RNA of <50 copies/mL at week 48	HIV-1 RNA of <50 copies/mL was maintained in 86.2% (25/29)—0.25 mg islatravir group, 90.0% (27/30)—0.75 mg islatravir group, 67.7% (21/31)—2.25 mg islatravir group, and 81.1% (73/90)—combined islatravir groups, compared with 80.6% (25/31) of participants who received DOR/3TC/TDF	Major AEs: Headache, diarrhoea, vomiting, and rash
	Johnson M., et al., 2019 [24]	Randomized, open-label, active-controlled, noninferiority trial	Adults with no history of virologic failure, HIV-1 RNA < 40 copies/mL, creatinine clearance ≥ 50 mL/min	Doravirine 100 mg, lamivudine 300 mg, TDF 300 mg	The proportion of participants with HIV-1 RNA < 50 copies/mL, comparison at week 48 in the DOR/3TC/TDF vs. week 24 in the Baseline Regimen	At week 24, participants with HIV-1 RNA < 50 copies/mL was 93.7% (419/447) in the DOR/3TC/TDF (maintained in 90.8% {406/447} at week 48) and 94.6% (211/223) in the Baseline Regimen. Switching to DOR/3TC/TDF is the same efficacy to continue the baseline regimen for 24 weeks	Major AEs: Nasopharyngitis and headache
	Kumar P., et al., 2021 [35]	Randomized, open-label, phase 3 trial	Adults with suppression HIV-1 for ≥6 months, no virologically failure	Doravirine 100 mg, lamivudine 300 mg, tenofovir 300 mg	The number of participants with plasma HIV-1 RNA < 50 copies/mL Comparison of DOR/3TC/TDF ISG at week 48 and Baseline Regimen DSG at week 24	HIV-1 RNA < 50 copies/mL was achieved in 80.1% of the immediate switch group, ISG (351/438), and 83.7% of the delayed switch group, DSG (175/209)	Major AEs: Nasopharyngitis, headache, and diarrhoea
Ibalizumab	Gathe JC., et al., 2021 [30]	Randomized, double-blind, placebo-controlled, 3-arm phase 2a study	HIV-1 Adults aged ≥18 years, failing their cART regimen	15 mg/kg ibalizumab IV infusions, placebo	Change from baseline viral load at week 24	The viral load had reduced at week 24 (−0.77 log10 for arm A and −1.19 log10 for arm B, versus −0.32 log10 for the placebo	Major AEs: Headache, diarrhoea, nausea, fatigue, and rash

## Data Availability

Data are contained within the article.

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
