# Peer review of "Efficacy and Safety of Two-Drug Regimens That Are Approved from 2018 to 2022 for the Treatment of Human Immunodeficiency Virus (HIV) Disease and Its Opportunistic Infections"

_microorganisms, 2023, doi:10.3390/microorganisms11061451_

Round 1
Reviewer 1 Report
The authors have done a good review of the subject. It is also written in a very clear way, making it very easy to read and understand.
Author Response
Thank you for the positive comments and feedback on our manuscript.
Reviewer 2 Report
A serious editing of the language is required.
Drug names are written in a confusing manner. First write names of the drugs than describe if given in combinations.
Information as to how the drugs are metabolized and why different combinations are contraindicated should be provided
Certain expressions require explanation, e.g., Ibalizumab-uiyk, why uiyk?
The section on “non-pharmacological management” is irrelevant.
The Table has been discussed and studies are referenced, and therefore, not needed.
You may provide a time-line of the discovery of the drugs in a graphical depiction.
Many confusing sentences here and there; they must be edited.
Author Response
Reviewer comment: A serious editing of the language is required.
Author’s response: The language of the manuscript has been revised and improved by the expert of a native English language-speaking colleague for more clarity of the information.
Reviewer comment: Drug names are written in a confusing manner. First write names of the drugs than describe if given in combinations.
Author’s response: The details of drugs are explained in the data extraction section of the material and methods for more clarity.
Reviewer comment: Information as to how the drugs are metabolized and why different combinations are contraindicated should be provided.
Author’s response: The details of each drug are included in the appropriate section.
Reviewer comment: Certain expressions require explanation, e.g., Ibalizumab-uiyk, why uiyk?
Author’s response: The drug name itself Ibalizumab-uiyk. There is no specific information provided by the manufacturer on what uiyk stands for. The details can be found at https://www.accessdata.fda.gov/drugsatfda_docs/label/2018/761065lbl.pdf
Reviewer comment: The section on “non-pharmacological management” is irrelevant.
Author’s response: The non-pharmacological section explains to the readers how non-pharmacological therapy supports the outcome of the patient from HIV infections. Concomitant administration of non-pharmacological therapy along with pharmacotherapy will speed up the recovery process and improve the QoL.
Reviewer comment: The Table has been discussed and studies are referenced, and therefore, not needed.
Author’s response: The table summarises the outcome of the systematic review of the two-drug regimen used in HIV infection. It can be deleted if the chief editor decided to remove it.
Reviewer comment: You may provide a time-line of the discovery of the drugs in a graphical depiction.
Author’s response: The timeline of the drug discovery is explained in short in the manuscript under the data extraction section of the materials and methods.
Reviewer comment: Comments on the Quality of English Language: Many confusing sentences here and there; they must be edited.
Author’s response: The Quality of the English Language is improved for more clarity, and the revised manuscript is reviewed and edited by native English speaker colleagues of our university.
Reviewer 3 Report
The manuscript is confusing and very difficult to follow, not just because of the English. It is incomplete and superficial. A few examples:
- The title announces an evaluation of TWO drug combinations for HIV and opportunistic infections. This creates false expectations as
o Only one two-drug combination, namely Cabuneva (Cabotegravir and Rilpivirine) is discussed. Mono-therapies of Fostemsavir and Ibaluzimab and tri-therapies with Doravirine are also discussed (hence NOT as TWO-drug combinations).
o There is no follow-up on treatment of opportunistic infections
- The efficacy data on Doravirine and Ibalizumab are very superficially described on p.6 (par 3.2.3 and 3.2.4), even without reference to viral suppression !
- About Fostemsavir
o on line 141: Fostemsavir (Rukobia) is an ARV drug used to cure HIV-1 infection.
o n line 229: … by prohibiting the virus from integrating into the human cell genetic makeup.
Clearly, both statements (“cure” and prohibiting from integrating”) are incorrect as Rukobia is a treatment and acts by inhibiting viral entry
- The information on resistance development to these drugs is very limited, while it is an important aspect for these “rescue” treatments.
- Etc…
Very difficult to read
Author Response
Reviewer comment: The manuscript is confusing and very difficult to follow, not just because of the English. It is incomplete and superficial. A few examples:
Author’s response: The contents are revised for more clarity of information, and more details are added as per the comments.
Reviewer comment: The title announces an evaluation of TWO drug combinations for HIV and opportunistic infections. This creates false expectations as
Only one two-drug combination, namely Cabuneva (Cabotegravir and Rilpivirine) is discussed. Mono-therapies of Fostemsavir and Ibaluzimab and tri-therapies with Doravirine are also discussed (hence NOT as TWO-drug combinations).
Author’s response: The two-drug regimen indicates not only a combination product, but it also refers to two drugs that are concomitantly used with other antiretroviral drugs for the treatment of HIV infections. This review included four drugs that are mainly indicated for concomitant use with other antiretroviral agents, as they can produce more effective responses when administered with other anti-HIV agents.
Reviewer comment: There is no follow-up on treatment of opportunistic infections
Author’s response: The drugs included in the review are not indicated for the treatment of opportunistic infections of HIV. Adding that treatment of opportunistic infections details will dilute the outcome of this review.
Reviewer comment: The efficacy data on Doravirine and Ibalizumab are very superficially described on p.6 (par 3.2.3 and 3.2.4), even without reference to viral suppression !
Author’s response: more details on the efficacy of Doravirine and Ibalizumab are included.
Reviewer comment: About Fostemsavir
o on line 141: Fostemsavir (Rukobia) is an ARV drug used to cure HIV-1 infection.
o n line 229: … by prohibiting the virus from integrating into the human cell genetic makeup.
Clearly, both statements (“cure” and prohibiting from integrating”) are incorrect as Rukobia is a treatment and acts by inhibiting viral entry
- The information on resistance development to these drugs is very limited, while it is an important aspect for these “rescue” treatments.
- Etc…
Author’s response: The mechanism of action of Fostemsavir (Rukobia) and the resistance developments are explained clearly in the appropriate section.